# Repeatability of Taste Recognition Threshold Measurements with *QUEST* and *Quick Yes–No*

**DOI:** 10.3390/nu12010024

**Published:** 2019-12-20

**Authors:** Richard Höchenberger, Kathrin Ohla

**Affiliations:** 1Institute of Neuroscience and Medicine (INM-3), Research Center Jülich, 52428 Jülich, Germany; richard.hoechenberger@gmail.com; 2Psychophysiology of Food Perception, German Institute of Human Nutrition Potsdam-Rehbrücke, 14558 Nuthetal, Germany

**Keywords:** sensitivity, taste, threshold, staircase, QUEST, bias, quick Yes–No (qYN)

## Abstract

Taste perception, although vital for nutrient sensing, has long been overlooked in sensory assessments. This can, at least in part, be attributed to challenges associated with the handling of liquid, perishable stimuli, but also with scarce efforts to optimize testing procedures to be more time-efficient. We have previously introduced an adaptive, QUEST-based procedure to measure taste sensitivity thresholds that was quicker than other existing approaches, yet similarly reliable. Despite its advantages, the QUEST procedure lacks experimental control of false alarms (i.e., response bias) and psychometric function slope. Variations of these parameters, however, may also influence the threshold estimate. This raises the question as to whether a procedure that simultaneously assesses threshold, false-alarm rate, and slope might be able to produce threshold estimates with higher repeatability, i.e., smaller variation between repeated measurements. Here, we compared the performance of QUEST with a method that allows measurement of false-alarm rates and slopes, *quick Yes–No* (qYN), in a test–retest design for citric acid, sodium chloride, quinine hydrochloride, and sucrose recognition thresholds. We used complementary measures of repeatability, namely test–retest correlations and coefficients of repeatability. Both threshold procedures yielded largely overlapping thresholds with good repeatability between measurements. Together the data suggest that participants used a conservative response criterion. Furthermore, we explored the link between taste sensitivity and taste liking or which we found, however, no clear association.

## 1. Introduction

The ability to taste is undoubtedly crucial for nutrient sensing. However, it is scarcely assessed in large cohorts, and precise and “practical” methods for the measurement of taste sensitivity are needed to get a better understanding of the extent to which taste shapes preferences and eating behavior.

The precise and reasonably quick measurement of sensory sensitivity has been a challenge for all senses. The relatively quick adaptation in gustation, however, increases the data collection burden enormously, as it requires long inter-stimulus intervals that cannot simply be countered with a reduction of the number of experimental trials. Using a Bayesian adaptive testing framework [1], we previously showed that taste sensitivity can be precisely and reliably measured in a fraction of the time needed with conventional, non-adaptive methods [2,3].

Bayesian adaptive methods such as QUEST [1] typically produce estimates of psychometric function parameters more quickly than conventional staircase procedures, especially if the number of possible stimuli is large. This is mainly due to two features specific to Bayesian methods: firstly, they can incorporate prior knowledge by assigning a probability to each individual parameter value; and, secondly, they rely on the entire response history of a participant to predict the “true” parameter value and select the next stimulus. Accordingly, on every trial, stimuli are selected such that the expected knowledge gain about the parameter (e.g., threshold) is maximized. This may lead to relatively strong intensity changes from one trial to the next, especially at the beginning of an experimental run, and allows the procedure to converge within fewer trials compared to a staircase.

When measuring thresholds, participants’ responses not only depend on their sensory sensitivity, but are also influenced by mental processes. Specifically, whether a participant will report the presence of a stimulus is influenced by their cognitive *response criterion*: if a *liberal* criterion is adopted, even weak, non-obvious stimuli will be reported; if the criterion, however, is *strict*, only stimuli that the participant believes are sufficiently strong are going to be reported. Generally, a liberal criterion leads to a higher false-alarm rate (FAR), which describes the proportion of “yes” responses to a blank, while a stricter criterion reduces the frequency of false alarms [4,5].

Although QUEST was designed for alternative forced-choice tasks, which are commonly thought to control the response criterion (cf. [6] for criticism of this view), the simplicity of the yes–no experiment (see [6] for a systematic review of adaptive procedures) has been a strong motivator to explore the suitability of QUEST in a yes–no design, and it has yielded good performance in the chemosensory domain [2,3,7]. In these studies, participants were instructed to be “conservative” in their response behavior, in an attempt to keep false-alarm rates low and constant across sessions. This is imperative because QUEST can only estimate a single parameter, such that, when the threshold is to be estimated, all other parameters defining the psychometric function, such as FAR, slope, and the lapse rate (i.e., the proportion of “no” responses to high-intensity, supra-threshold stimuli) need to be set to a fixed value a priori. While the lapse rate can safely be assumed to be low in taste threshold testing, provided that sufficiently long inter-stimulus intervals are used and participants thoroughly rinse their mouth between trials, FAR may vary between repeated measures despite the instructions. Slope, on the other hand, determines how well participants can detect intensity differences between stimuli, and as such can also serve as an (implicit) measure of the reliability of the threshold estimate. Similarly to how thresholds differ between participants and, obviously, tastants, it is known from other sensory systems that “slopes are different for different stimuli, and this can lead to misleading results if slope is ignored” [6].

Here, we set out to address these concerns, and tested whether FAR and slope indeed vary between tastants and sessions, and whether the measurement of these parameters helps improve the repeatability of taste threshold measurements. To this end, we derived a test–retest design in which we employed two Bayesian procedures, the previously used QUEST, which only measures threshold, and qYN [8], which assesses the sensory threshold, FAR, and psychometric function slope. Additionally, we assessed how much participants liked different tastes and how often they consumed foods and beverages that have a certain taste to test whether these aspects are related to taste sensitivity.

## 2. Materials and Methods

### 2.1. Participants

Forty-one participants (34 women; mean age M=30.1, standard deviation SD = 11.4, and range 18–64 years) participated in the study and received compensatory payment. Their weight was within the normal range according to the body mass index (BMI; M=22.5, SD = 2.6). Four participants were smokers. Exclusion criteria were self-reported taste and smell disorders; current or recent oral, nasal, or sinus infections; pregnancy; recent (during the last six months) childbirth; metabolic disease; and recent (during the last three months) weight change exceeding 10 kg. No participants were excluded; however, the samples for the different analyses vary slightly because individual data were missing. The age and sex structure for each sub-sample can be obtained from the associated data file (see Section 6). The study conformed to the revised Declaration of Helsinki and was approved by the ethical board of the German Society of Psychology (DGPs).

### 2.2. Procedure

#### 2.2.1. Experimental Sessions

Participants were invited to four experimental sessions that lasted 1 h each: a Test and a Retest session for each of the two threshold procedures. To ensure similar testing conditions across sessions, participants were instructed to refrain from eating and drinking anything but water 30 min before visiting the laboratory. Further, the sessions were scheduled at approximately the same time of day, and within 10 days (inter-session interval: M=2.3, SD = 2.0, range 1–10 days).

At the beginning of the first session, participants completed a screening questionnaire, the Dutch Eating Behavior Questionnaire (DEBQ) [9], and rated how much they like and how often they typically consume salty, sour, sweet, and bitter foods. Following the ratings and in each subsequent session, taste recognition thresholds for citric acid (sour), sodium chloride (salty), quinine hydrochloride (bitter), and sucrose (sweet) were measured using either of the two algorithms, QUEST or qYN, described below. The order of tastants was balanced across participants and kept constant for Test and Retest within each participant. The order of algorithms was balanced across participants.

#### 2.2.2. Eating Behavior, Taste Liking, and Food Consumption

Eating style was measured with the DEBQ [9], which assesses—along three dimensions—the degree of restrained, emotional, and external eating behavior. Participants could choose between the German and English version; seven participants completed the English version. The questionnaire data of one participant (female, 27 years old) are missing.

Furthermore, participants rated how much they liked salty, sour, sweet, and bitter foods and beverages on separate, horizontal five-point Likert scale anchored with 1 (not at all) and 5 (extremely).

They furthermore provided the frequency at which they typically consume salty, sour, sweet, and bitter foods and beverages on a scale with seven options: daily (score 7), 4–6 times per week (6), 2–3 times per week (5), once per week (4), 2–3 times per month (3), once per month (2), and less than once per month (1). Specifically, they provided ratings for the following items: sweet, sour, and bitter beverages; sweet, sour, and bitter fruits and vegetables; sweet cake/candy; salty snacks; and added salt. Rating for each taste quality were averaged for further analysis. The ratings were assessed in paper-and-pencil format. Ratings from four participants (all female, 21–27 years old) are missing.

#### 2.2.3. Taste Recognition Thresholds

##### Procedure

Participants were seated comfortably in the laboratory and blindfolded to reduce distraction and improve focus. At the beginning of each measurement, they were told which taste would be tested next. At the beginning of each trial they were asked to stick out the tongue and received the stimulus. Participants were required to indicate whether they recognized the taste by nodding (“yes”) or shaking their head (“no”) while they kept their tongue extended. Immediately after the response, the experimenter entered it into the computer, and participants rinsed their mouth with deionized water. Participants received no feedback as to their performance during the experiment. The interval between consecutive stimuli was approximately 30 s.

##### Taste Stimuli

Tastants were prepared by diluting prototypical chemicals that are known to elicit a clear taste perception in deionized water: citric acid (sour; molar mass M=192.12 g/mol), sodium chloride (salty; M=58.44 g/mol), quinine hydrochloride (bitter; M=396.91 g/mol), and sucrose (sweet; M=342.30 g/mol). All chemicals were produced by Sigma-Aldrich and purchased from Merck KGaA, Darmstadt, Germany.

Based on previous studies [2,3], we used the following sets of different concentrations that were equidistantly spaced on a decadic logarithmic grid for each tastant: citric acid, 0.015 mM to 46.846 mM (14 $\log_{10}$ steps; step width: 0.269); sodium chloride, 0.342 mM to 342.231 mM (12 $\log_{10}$ steps; step width: 0.273); quinine hydrochloride, 0.077 × 10^−3^ mM to 3.131 mM (21 $\log_{10}$ steps; step width: 0.230); and sucrose, 0.073 mM to 584.283 mM (14 $\log_{10}$ steps; step width: 0.300).

Taste solutions were stored refrigerated at 4 ${}^{\circ }C$ for a maximum duration of seven days in glass bottles. During testing, they were sprayed manually by the experimenter to the anterior half of the tongue using a conventional spray head that was screwed onto the glass bottle and released approximately 0.2
mL of mist.

##### Psychometric Functions

The QUEST implementation we used assumes a psychometric function in which the proportion of “yes” responses to a stimulus is given by
$$\Psi_{\mathrm{yes}}\left(c\right)=\delta \gamma +\left(1-\delta \right)\left[1-\left(1-\gamma \right)\exp\left(-10^{\beta (c-\tau +\epsilon )}\right)\right].$$

Here, *c* is the concentration of a given tastant and τ is the threshold concentration one wishes to estimate. The parameter ϵ shifts the psychometric function such that threshold performance (proportion of “yes” responses) equals a predefined value. ϵ was determined automatically by our software based on the desired threshold performance. The parametrization of the procedure was identical to the one used by Höchenberger and Ohla [2]: We defined “threshold” as the concentration with an expected proportion of 80% “yes” responses ($\Psi_{\mathrm{yes}}\left(c\right)=0.80$); the prior probabilities for the threshold parameter were given by a normal distribution with a standard deviation of 20, centered on the starting concentration of the respective tastant, i.e., the concentration used in the first trial. All other parameters were fixed a priori: slope, β, to 3.5; and both the false-alarm and lapse rate, γ and δ, to 0.01. Internally, QUEST works with an abstract “’intensity grid”, whose granularity we set to 0.01 as well.

In quick Yes–No [8], the psychometric function describing the proportion of “yes” responses for a given concentration is
$$\Psi_{\mathrm{yes}}\left(c\right)=\epsilon +\left(1-\epsilon \right)\left[1-\Phi \left(\lambda -d^{\prime }\left(c\right)\right)\right]$$
with Φ being the cumulative normal distribution with a mean of 0 and a standard deviation of 1. ϵ is the lapse rate, which describes how frequently stimuli of a high intensity are not recognized. The decision criterion, λ, is related to the FAR via the percent point function of the normal distribution: $\lambda =\Phi^{-1}\left(1-\mathrm{FAR}\right)$ [8]. $\Psi_{\mathrm{yes}}$ depends on the sensitivity function, $d^{\prime }$, given by [8]
$$d^{\prime }\left(c\right)=\frac{\beta {\left(c/\tau \right)}^{\gamma }}{\sqrt{\left(\beta^{2}-1\right)+{\left(c/\tau \right)}^{2\gamma }}}.$$

τ is the “threshold intensity”, which here is defined as the tastant concentration corresponding to a sensitivity of $d^{\prime }=1$. β defines the upper asymptote and γ the slope. In this study, we estimated threshold, slope, and decision criterion. Consequently, the parameter space was a three-dimensional grid. Different ranges of τ were used for each tastant. To achieve a finer granularity, additional (virtual) concentration steps were inserted halfway between the existing (physical) concentration steps, producing 27 values for citric acid and sucrose, 23 for sodium chloride, and 35 for quinine hydrochloride. For γ, we used 10 values in the interval [0.5,3.0], evenly spaced on a decadic logarithmic grid. For λ, we used eight evenly spaced values in the interval [0.75,2.50], corresponding to FARs of [22.7%,<1.0%]. We assumed no prior knowledge regarding the “true” parameter values, and, hence, used an “uninformative” prior that assigned the same probability to all possible parameter value combinations. β was fixed at 5.0 [8], and ϵ was set to 0.

##### Stimulus Selection

The concentration presented in the first trial for each tastant was predefined such that it would be supra-threshold for most participants, in order to familiarize them with the particular tastant as testing commenced (citric acid: 7.328
mm; sodium chloride: 97.469
mm; quinine hydrochloride: 0.077
mm; and sucrose: 73.509
mm). For subsequent trials, the QUEST and qYN procedures proposed the stimulus concentration to present based on response behavior in all previous trials. While QUEST aims to place stimuli at threshold concentration, qYN—trying to estimate *three* parameters at once—typically also suggests to present stimulus concentrations slightly above and below threshold to measure slope, and at very low concentrations to estimate the FAR.

As both algorithms internally worked with smaller, virtual concentration steps, they would sometimes propose concentrations that were not physically available. In this case, our computer program selected the concentration closest to the proposed one, and informed the algorithm about the actually used concentration. In QUEST, we added an additional rule: whenever the algorithm proposed to present the same concentration on two consecutive trials, we increased the concentration in the second trial by one step if the participant had responded “no” in the previous trial, and we decreased the concentration by one step if the response in the previous trial was “yes”, thus introducing a little more variability to avoid repetitive presentation of the same concentration on multiple consecutive trials, which we felt could have been more tiring for participants. In qYN, we did not add such a rule, as the algorithm itself introduced somewhat abrupt concentration changes once in a while in order to determine FAR and slope.

##### Taste Recognition Termination

qYN experimental runs always ended after 20 trials. For QUEST, we employed the same termination criterion as in a previous study [2]: after more than 10 trials had been performed, we checked after each trial whether the 90% confidence interval of the threshold estimate was smaller than half a concentration step; if that was the case, the experimental run was finished. Otherwise, a maximum of 20 trials was performed.

##### Parameter Estimates

To retrieve the final threshold estimate, we calculated the mean of the concentrations weighted by the posterior distribution (QUEST) or by the marginal posterior distribution (qYN), respectively. This measure had been shown to produce an unbiased estimate of threshold in QUEST, as opposed to other metrics [10]. We limited the values of the threshold estimates to the range of stimulus concentrations used in the present study, as QUEST could—in rare cases—produce thresholds outside of this range for extremely sensitive or insensitive participants. This was the case for a single participant where QUEST produced one quinine hydrochloride threshold above the highest available concentration.

For qYN, FAR and slope were calculated as the mean of the respective parameter space weighted by the corresponding marginal posterior.

### 2.3. Analysis

The significance level α was set to 0.05 a priori for all statistical tests. Greenhouse–Geisser correction was applied for violation of sphericity in repeated measures analysis of variance (rmANOVA); uncorrected degrees of freedom and corrected *p*-values are reported in this case.

#### 2.3.1. Ratings

Ratings for taste liking and frequency of consumption were submitted to separate one-factorial rmANOVA with four levels (sour, salty, sweet, and bitter). To quantify the potential link between taste liking and sensitivity, we computed Spearman’s correlation coefficients. For this, the average of all threshold estimates from QUEST and qYN for a given participant was used as robust measure for taste sensitivity. Data from only 37 participants are reported for liking and frequency of consumption because four participants did not complete the ratings.

#### 2.3.2. Taste Recognition Data Cleaning

Out of the 656 obtained thresholds (41 participants × 4 tastants × 2 procedures × 2 sessions), 41 (6.3%; 17 QUEST and 24 qYN) were lost during the transfer of electronic data, yielding 615 datasets. In many instances, only data from a single session were lost; we only used the measurements for which data from *both* sessions were available. The resulting 590 datasets entered analysis (90% of the total data; 302 QUEST and 288 qYN runs). All included and omitted datasets are available online (see Section 6 for details).

#### 2.3.3. Test–Retest Reliability

##### Threshold

We first submitted the threshold estimates to separate rmANOVAs for each tastant with the factors *procedure* (QUEST and qYN) and *session* (Test and Retest) to test for systematic differences between procedures and measurement repetitions.

We then calculated Spearman’s rank correlation, ρ, for each tastant to quantify the monotonic relationship between the measurement results in both sessions.

Correlation analysis does not necessarily provide a good indication of absolute *repeatability* of an experiment, as “[the] correlation coefficient is a reflection of how closely a set of paired observations follow a straight line, regardless of the slope of the line” [11], and it also disregards systematic changes between measurements [11,12], such as a constant offset. We, therefore, conducted an additional analysis that focused on the *differences* between measurements. For each procedure separately, we first calculated the difference between Test and Retest estimates for all participants. Then, we calculated the standard deviation of these differences, $s_{d}$, and derived a coefficient of repeatability (CR), $\mathrm{CR}=1.96\times s_{d}$ [13]. Note that it has been suggested [11,12,14] to calculate CR as $\sqrt{2}\times 1.96\;s_{w}$, where $s_{w}$ is the *within-participant standard deviation*, i.e., the square-root of the averaged within-participant variances of measurement repetitions. We found that, with our data, this approach produced very similar results (deviating only in the second decimal place) to the simpler formula $1.96\times s_{d}$, which directly and intuitively corresponds to the 95% limits of agreement in the Bland–Altman plots. Therefore, we elected to follow this simpler approach, and report CRs based on the SD of measurement differences between sessions here. The number 1.96 is a *z*-score and corresponds to the 97.5% quantile of the normal distribution. If a participant were to be measured repeatedly using the same procedure, we would then expect 95% of the absolute measurement differences not to exceed CR. This provides a straightforward, single-number representation of the magnitude of measurement variation to expect. Lastly, we calculated the mean of the differences between sessions, $\bar{d}$, and estimated the *95% limits of agreement* (LoA) as $\mathrm{LoA}=\bar{d}\pm \mathrm{CR}$ [13]. These limits correspond to the 95% confidence interval of the differences, and, consequently, narrower LoAs suggest better measurement repeatability.

Because the calculations of the mean difference and LoAs are based on an experimental sample, they are estimates that naturally have a certain amount of uncertainty associated with them. We therefore also derived 95% confidence intervals (CIs) of these estimates. The mean difference was assumed to be normally distributed with mean $\bar{d}$ and SD $s_{d}/\sqrt{n}$ [14] (with the number of paired samples, *n*), and hence the CI corresponded to the 2.5% and 97.5% quantiles of this distribution. CIs of the LoAs were calculated via the “exact paired” method [15].

For visual comparison, we plotted the differences between Test and Retest over the mean of both measurements (which serves as our best estimate of the “true” value) and added the mean difference $\bar{d}$ and LoAs as horizontal lines, producing so-called *Bland–Altman* or *Tukey mean difference plots*. These plots allow for a quick and straightforward inspection of measurement differences, exposing systematic biases ($\bar{d}\neq 0$), the degree of measurement differences, and their variability.

##### False-Alarm Rate (FAR)

The decision criterion parameter λ, which was only estimated by qYN, was first transformed to a proportion of false alarms: FAR=1−Φ(λ). The FARs were submitted to an rmANOVA with the factors *tastant* (citric acid, sodium chloride, quinine hydrochloride, and sucrose) and *session* (Test and Retest) to investigate whether different tastants and measurement repetitions affected FARs differentially.

We then constructed Bland–Altman plots as described above. The plots revealed that the difference between both sessions changed with the magnitude of the session mean. However, the calculation of CRs and LoAs requires the differences between sessions to be approximately normally distributed to work correctly. If that is not the case, a logarithmic transformation can be carried out prior to analysis [12]. Here, we elected to use the decadic logarithm, $\log_{10}$. After the data had been transformed, $\bar{d}$, CR, LoAs, and CIs were calculated following the same procedure as described above for thresholds. To plot the results that were calculated in log space, $y_{\log}$, in their original coordinate system for intuitive visual assessment, a back-transformation must be applied [16]: $y_{\mathrm{back}}=2M\;\left(10^{y_{\log}}-1\right)/\left(10^{y_{\log}}+1\right)$, where *M* is a given value on the abscissa (i.e., any given session mean). The result of the back-transformation, therefore, describes a line, not a single value, in the original coordinate system. In other words, the value of the back-transformed parameter, $y_{\mathrm{back}}$, is conditional on *M*. The generated plots can be interpreted in the same way as a Bland–Altman plot that did not undergo a logarithmic transformation.

##### Slope

Similarly to FARs, the slopes estimated by qYN were analyzed using rmANOVA with the factors *tastant* and *session*, and Bland–Altman plots, CRs, LoAs, and their CIs were derived as described above for the threshold data.

##### Relationship between FAR and Slope

As the FAR determines the location of the lower asymptote, changes in FAR necessarily lead to changes in steepness of the psychometric function if the threshold is assumed to remain unchanged: as FAR increases, the slope must decrease, and vice versa. We thus calculated Spearman’s rank correlation between FARs and slopes, pooled across all tastants, to test whether covariance of both variables could explain their fluctuations between sessions.

#### 2.3.4. Software

Stimulus presentation and data collection were guided by a Python computer program based on PsychoPy 1.85.4 [17] on Windows 7 (Microsoft Corp., Redmond, WA/USA). Statistical analyses were carried out with JASP 0.11.1 (https://jasp-stats.org/) and pingouin 0.2.9 [18]. CIs for the Bland–Altman plots were calculated via pyCompare (https://github.com/jaketmp/pyCompare). Plots were created using matplotlib 3.1.1 (https://matplotlib.org) and seaborn 0.9.0 (https://seaborn.pydata.org).

## 3. Results

### 3.1. Eating Behavior, Taste Liking, and Food Consumption

All measurement outcomes are summarized in Table 1.

The DEBQ eating style scores conformed well with recent norm data [19] and participants’ eating behavior can hence be considered normal.

The liking of different tastes varied significantly ($F_{3,36}=33.68$, p<0.001, $\eta_{p}^{2}=0.483$), as expected, with bitter taste being significantly less liked than salty ($t_{36}=-6.95$, $p_{\mathrm{holm}}<0.001$), sour ($t_{36}=-7.17$, $p_{\mathrm{holm}}<0.001$), and sweet ($t_{36}=-9.65$, $p_{\mathrm{holm}}<0.001$); and with sweet being significantly more liked than salty ($t_{36}=2.77$, $p_{\mathrm{holm}}=0.018$) and sour ($t_{36}=4.2$, $p_{\mathrm{holm}}<0.001$). Accordingly, scores (min = 1, max = 5) were highest for sweet, followed by salty, sour, and bitter. In contrast to sweet, sour, and salty, which were generally liked (as indicated by ratings larger than 3), bitter was clearly disliked (as indicated by a rating smaller than 3). The liking of sour taste correlated significantly with the taste threshold for sucrose (ρ=0.403, p=0.013) and marginally for citric acid (ρ=0.305, p=0.066). No further correlations between taste liking and thresholds were found (all p>0.19).

Participants reported to consume food and beverages of the different tastes only approximately once per week (score = 4) for all taste qualities. Because these reported mean frequencies raise doubts as to their validity, no further statistical analyses were conducted.

### 3.2. Taste Recognition

The distributions of threshold estimates, averaged across both sessions and split by procedure, are shown in Figure 1. The rmANOVAs revealed a main effect of *procedure*—indicating that threshold estimates were systematically lower for qYN compared to QUEST—for citric acid ($F_{1,32}=17.475$, p<0.001, $\eta_{p}^{2}=0.353$), sodium chloride ($F_{1,32}=44.728$, p<0.001, $\eta_{p}^{2}=0.583$), and sucrose ($F_{1,31}=11.198$, p<0.01, $\eta_{p}^{2}=0.265$), but not for quinine hydrochloride ($F_{1,33}=3.241$, p=0.08, $\eta_{p}^{2}=0.089$). Thresholds did not differ significantly between sessions for sodium chloride ($F_{1,32}=0.342$, p=0.56), quinine hydrochloride ($F_{1,33}=1.195$, p=0.28), and sucrose ($F_{1,31}=0.219$, p=0.64); however, we found a main effect of *session* for citric acid ($F_{1,32}=4.492$,p=0.042, $\eta_{p}^{2}=0.123$). No interactions between *procedure* and *session* were found (all p>0.13). The threshold estimates, their respective minimum and maximum values, and their standard deviations are listed in Table 2. Graphical representations of the psychometric functions, generated based on the mean parameter estimates (i.e., averaged across participants), are shown in Figure 2. Figures of individual psychometric functions are available online; see Section 6 for details.

On average, QUEST needed 14.6 trials—corresponding to 6:50 min—to converge to a threshold. For qYN, the duration of an experimental run was always 9:30 min, as the number of trials was fixed to 20.

#### 3.2.1. Threshold Repeatability

Test and Retest threshold estimates correlated significantly for all tastants in both procedures: QUEST (citric acid: $\rho_{35}=0.62$; sodium chloride: $\rho_{36}=0.63$; quinine hydrochloride: $\rho_{36}=0.80$; and sucrose: $\rho_{36}=0.67$; all p<0.01) and qYN (citric acid: $\rho_{35}=0.71$; sodium chloride: $\rho_{34}=0.60$; quinine hydrochloride: $\rho_{34}=0.76$; and sucrose: $\rho_{33}=0.76$; all p<0.01) (see Figure 3). To gain a better understanding of the nature of the individual differences between Test and Retest thresholds, we constructed Bland–Altman plots for each tastant in both procedures (Figure 4). The 95% confidence intervals of the mean differences always included 0, providing no evidence of systematic differences between sessions. The respective CRs for QUEST and qYN were, in $\log_{10}$ mM: 0.97 and 0.84 for citric acid (corresponding to 3.6 and 3.1 concentration steps), 0.98 and 0.95 for sodium chloride (3.6 and 3.5 steps), 1.07 and 1.29 for quinine hydrochloride (4.7 and 5.6 steps), and 1.10 and 1.03 for sucrose (3.7 and 3.4 steps). The mean CR across all tastants was 3.90 steps for both procedures. 95% of all Test and Retest differences fall into this range.

#### 3.2.2. False-Alarm Rates and Psychometric Function Slopes

To investigate potential differences between tastants and across sessions in estimated FARs and psychometric function slopes, we conducted two rmANOVAs with the factors *tastant* and *session*, with FAR and slope as the respective dependent variables.

For FARs, we found no main effects and no interaction of the factors (all p>0.25), indicating there was no evidence that response criteria would systematically vary with tastants or across sessions. We therefore decided to pool all data points, yielding a mean FAR of 0.059 (SD = 0.024) spanning across a relatively wide range (0.017–0.216). However, only a single of the 37 participants showed FARs >0.10 in *all* measurements. The Bland–Altman plot revealed an increasing variability of the differences between Test and Retest as mean FARs increased (Figure 5). This finding can be interpreted such that some participants would expose a relatively high FAR in one session, but a small FAR in the other. For the majority of participants, however, FARs varied within a relatively narrow range. Because the FAR differences between sessions were obviously not normally distributed, we $\log_{10}$-transformed the data for the calculations of mean difference, CR, LoAs, and the corresponding CIs. The results were then back-transformed to the original scale of the data [16]. As can be seen in Figure 5, the back-transformation does not yield a single CR value, but *a line* describing how CR changes as a function of the mean of Test and Retest. The resulting CR was 0.969×sessionmean.

For $d^{\prime }$ slopes, we found a significant main effect of *tastant*, albeit with a small effect size ($F_{3,96}=3.05$, p=0.04, $\eta_{p}^{2}=0.09$). This finding suggests that the ability to discriminate between stimuli of adjacent concentration steps systematically shifted with tastants. A post-hoc *t*-test revealed that this effect was driven by a significant difference between sodium chloride and sucrose slopes ($t_{33}=2.857$, $p_{\mathrm{holm}}=0.045$, d=0.497). There was no effect of *session* and no interaction between the factors (both p>0.19). Mean slopes pooled across sessions were 1.49 (SD = 0.53) for citric acid, 1.69 (SD = 0.60) for sodium chloride, 1.46 (SD = 0.57) for quinine hydrochloride, and 1.45 (SD = 0.46) for sucrose. Bland–Altman plots for all tastants are shown in Figure 6. In agreement with the rmANOVA results, differences between sessions did not significantly deviate from zero, as indicated by the confidence intervals spanning across 0. CRs ranged from 0.99 (sucrose) to 1.32 (sodium chloride), which is large, considering the mean slopes.

There was a significant correlation between FARs and slopes ($\rho_{286}=-0.749$, p<0.001), indicating that higher FARs were associated with reduced steepness of the $d^{\prime }$ sensitivity function.

## 4. Discussion

Using two Bayesian procedures, based on QUEST [1] and qYN [8], we explored the impact of false-alarm rate and psychometric function slope estimation on the precision and accuracy of taste sensitivity measurements.

### 4.1. Taste Recognition Thresholds

The comparison between session means for each procedure showed slightly, but systematically higher thresholds for QUEST compared to qYN for citric acid, sodium chloride, and sucrose; the difference was not significant for quinine hydrochloride, which we believe can be attributed to the larger variability of bitter threshold measurements compared to the other taste qualities. These differences were expected because the exact shape and parameterization of the psychometric functions and the definition of “threshold” differed between the procedures.

### 4.2. Threshold Repeatability

Both QUEST and qYN thresholds showed good monotonic relationships across sessions, as indicated by test–retest correlations ranging from ρ=0.62 to ρ=0.86 for QUEST and from ρ=0.60 to ρ=0.76 for qYN. No procedure produced consistently higher correlations than the other: qYN correlations were stronger for citric acid and sucrose, while QUEST showed higher correlations for sodium chloride and quinine hydrochloride. These correlations compare very well with a past applications of the QUEST method (r=0.59–0.83 [2]), with a modified Harris–Kalmus (r=0.70–0.77 [20]), and a forced-choice staircase procedure (r=0.76–0.86 [21]). The latter two approaches required notably longer testing times than QUEST. Other researchers have also observed much smaller correlations, for example with the relatively quick three-drop method (r=0.36–0.61 [22]) and with taste strips (r=0.34–0.56 [22]).

The observed correlation coefficients do not, however, account for systematic changes occurring between sessions, and they do not necessarily honor the spread of the data and the slope associated with their relationship [11]. Therefore, we (a) created Bland–Altman plots [12,14,23] to visualize the distributions of differences, and (b) calculated coefficients of repeatability (CR) [13] as an estimate of the expected measurement differences between sessions in an individual participant. We found that thresholds did not vary systematically across sessions. qYN produced smaller CRs than QUEST—indicating better agreement between measurement repetitions–for all tastants except quinine hydrochloride. The CR averaged across tastants was identical for both procedures at an equivalent of 3.90 concentration steps. Using a QUEST procedure in a 3-AFC task to estimate smell thresholds [7], we previously observed a CR corresponding to approximately 5.3 concentration steps on a $\log_{10}$ grid with a step width of 0.300, which is similar to the step width used here for sucrose, and larger than the step width for all other tastants. Neglecting the task differences (yes–no in the present study versus 3-AFC in [7]), the QUEST procedure seems to perform better for taste than for smell measurements.

We would like to emphasize a discrepancy between correlation coefficients and repeatability (measured as CRs). In QUEST, repeatability was *highest* for citric acid, yet the corresponding correlation was the *lowest* of all of the four tastants. Similarly, in qYN, repeatability was *lowest* for quinine hydrochloride and sucrose, yet the corresponding correlations were *highest*.

Correlation coefficients are commonly adopted to quantify repeatability in the chemical senses literature, and should therefore be calculated to enable comparisons with previously published studies. However, thorough examination of the correlated data is required to ensure that the conclusions drawn from these analyses are not inadvertently erroneous. We, therefore, suggest to always visualize the data in a scatter plot and the identity line to uncover systematic changes between measurements, which can occur even if the data points are highly correlated. To better understand the spread and pattern of measurement differences, Bland–Altman plots and coefficients of repeatability (CR) should be derived [11,12,13,14,23]. CRs indicate the magnitude of differences to expect when applying an experimental procedure repeatedly, and help guide the decision whether that procedure is suitable for a particular investigation, e.g., a clinical assessment.

### 4.3. False-Alarm Rates and $d^{\prime }$ Slopes

False-alarm rates (FARs) and $d^{\prime }$ slopes were assessed in qYN runs. FARs did not differ between tastants, indicating that participants’ response criterion was not taste-specific. Furthermore, FARs were generally low, suggesting that most participants complied with the instruction to be conservative in their response behavior. This finding was further substantiated through inspection of FAR differences between Test and Retest, which revealed little variation in most participants. However, session differences grew as the magnitude of FARs increased, leading to a relatively high FAR in one session, but a much smaller FAR in the other.

Slopes of the sensitivity function ($d^{\prime }$) only differed between sodium chloride and sucrose. CRs for slopes were relatively large. While this parameter of the psychometric functions is known to be notoriously difficult to estimate, especially if the number of trials is low [8], slope and FAR are also directly linked: for a given sensitivity threshold, a lower FAR will lead to a steeper psychometric function while a higher FAR demands a reduction in steepness. We found evidence for this dependence, as FAR and slope were strongly negatively correlated.

Differences in FAR between sessions can be interpreted such that participants followed different cognitive strategies in the two measurements, i.e., they changed their response criterion. These changes in FAR, then, would inevitably affect the slope as well and vice versa.Inspection of the trial sequences of experimental runs with the highest FARs (in the 90th percentile and above) revealed that, here, participants had indeed responded “yes” to stimuli of very low concentrations that were clearly below threshold. This shows that it is more likely that FAR changes lead to slope adjustments, than vice versa. Overall, the results support our premise in the QUEST procedure that FARs are low and stable in the majority of participants.

### 4.4. Measurement Duration

On average, QUEST finished 2.40 min more quickly than qYN, thanks to its dynamic termination criterion that ends the experimental run when the confidence interval around the threshold estimate reaches a predefined low limit. qYN, on the other hand, always completes 20 trials because no termination criterion was set to ensure sufficient data for the simultaneous estimation of the three parameters, threshold, FAR, and slope. Whether the amount of testing time required for qYN could be reduced by employing a similar dynamic stopping criterion needs to be tested in future studies.

### 4.5. Taste Sensitivity, Taste Liking, and Food Preference

In addition to taste sensitivity, we assessed taste liking. The data revealed no clear link between the two, with the exception of a positive correlation between sour liking and sucrose and—by trend—sour threshold. Accordingly, sour liking was higher in participants with lower sucrose and citric acid sensitivity (higher threshold). Whether this association has the potential to shape taste and food preferences or even food intake remains unanswered in the present study, as the reported food frequencies appeared to be unrealistically low and were hence not analyzed. A recent study, however, in which food and beverage consumption was thoroughly assessed [24], showed clear associations between salty, sweet, and bitter taste sensitivity and the intake of foods and beverages with these taste qualities. In this study, higher sensitivity was linked with lower food intake (or vice versa). No such link was found for sour taste, however, leaving it open whether taste sensitivity should be considered a general predictor for food intake behavior. The observation that a low sour taste liking and poor sour taste abilities improved through the course of a weight loss intervention in obese children indicates that dietary changes may influence preference as well as taste function, at least to some extent [25]. The latter data provide a glimpse into the complex and potentially reciprocal interplay of taste function, taste preference, and food intake and a review on the determinants of fruit and vegetable consumption revealed that other factors such as age, gender, socio-economic status, preferences, parental intake, and availability play a major role as well [26], thereby corroborating the multifaceted nature of food intake behavior and highlighting the need for further studies.

## 5. Conclusions

We compared the repeatability of taste recognition threshold estimates produced by two adaptive procedures, QUEST and qYN, for citric acid, sodium chloride, quinine hydrochloride, and sucrose. Both procedures select stimulus concentrations such that—based on a participant’s entire response history—the expected information gain about the true parameter(s) of a psychometric function is maximized. While QUEST only assesses the threshold, qYN also adjusts FAR and slope. Our analysis consisted of the widely adopted calculation of correlation coefficients between repeated measurements, and the estimation of coefficients of repeatability (CR) to assess the expected difference between two measurements of the same participant. The magnitudes of test–retest correlations were generally good and not clearly in favor of either threshold procedure. The CRs, however, revealed slightly better repeatability of qYN thresholds for citric acid, sodium chloride, and sucrose, compared to QUEST. The good agreement between both methods together with the low FARs observed in qYN suggest that, overall, participants applied a conservative response criterion as instructed.

## 6. Data and Software Availability

The data reported in this paper along with graphical representations of individual threshold runs and psychometric functions are available for download from https://doi.org/10.5281/zenodo.3540534.

## Figures and Tables

**Figure 1 nutrients-12-00024-f001:**
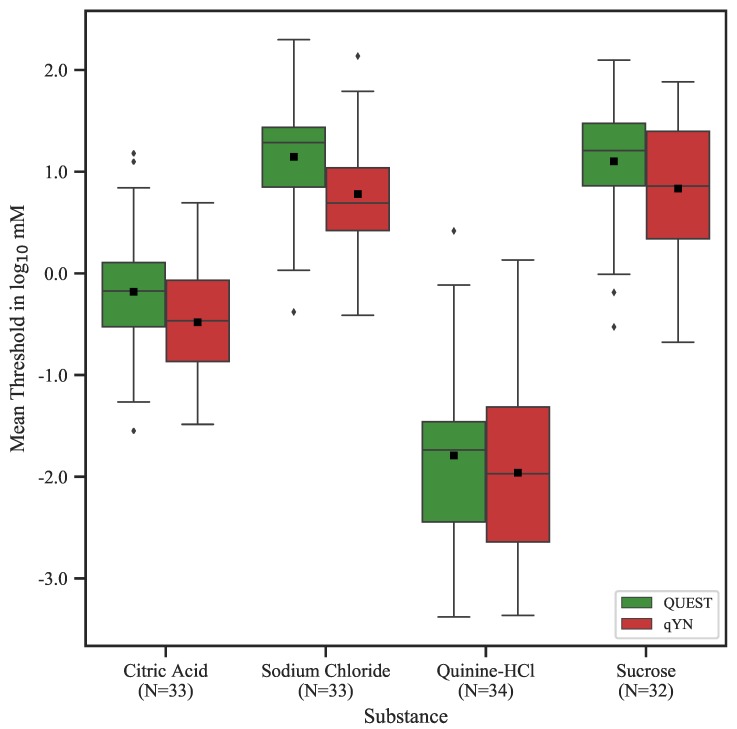
Distributions of the means of Test and Retest threshold estimates, split by tastant and procedure. Squares indicate the mean, and whiskers correspond to 1.5× inter-quartile range. Only participants for whom threshold data for both sessions and procedures were available are included; the number of participants is given below the respective abscissa labels for each tastant.

**Figure 2 nutrients-12-00024-f002:**
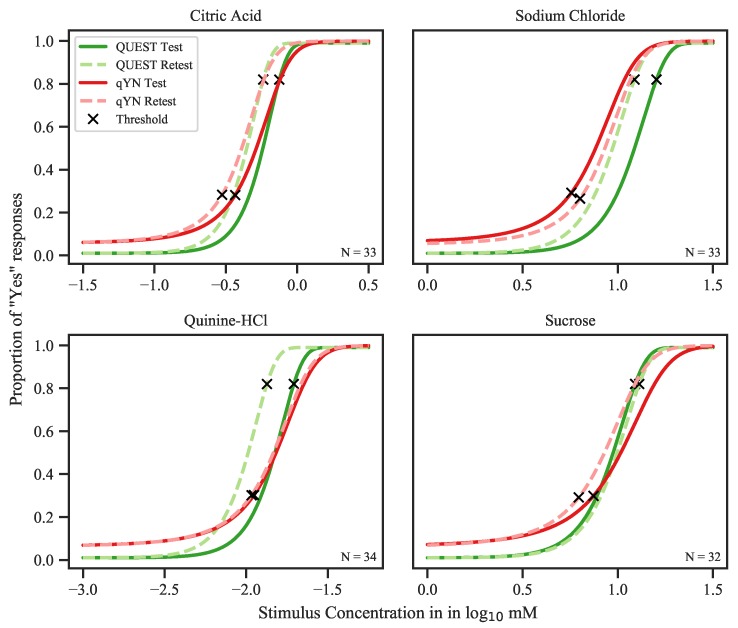
Psychometric functions based on the averaged parameter estimates. Mean thresholds are depicted as crosses. It becomes apparent how the definition of “threshold” differs between QUEST and qYN: while threshold in QUEST is solely based on the proportion of “yes” responses, qYN uses a threshold based on the sensitivity function $d^{\prime }$, which also takes false alarms into account. Only data from participants for whom threshold data of both sessions and procedures were available are included.

**Figure 3 nutrients-12-00024-f003:**
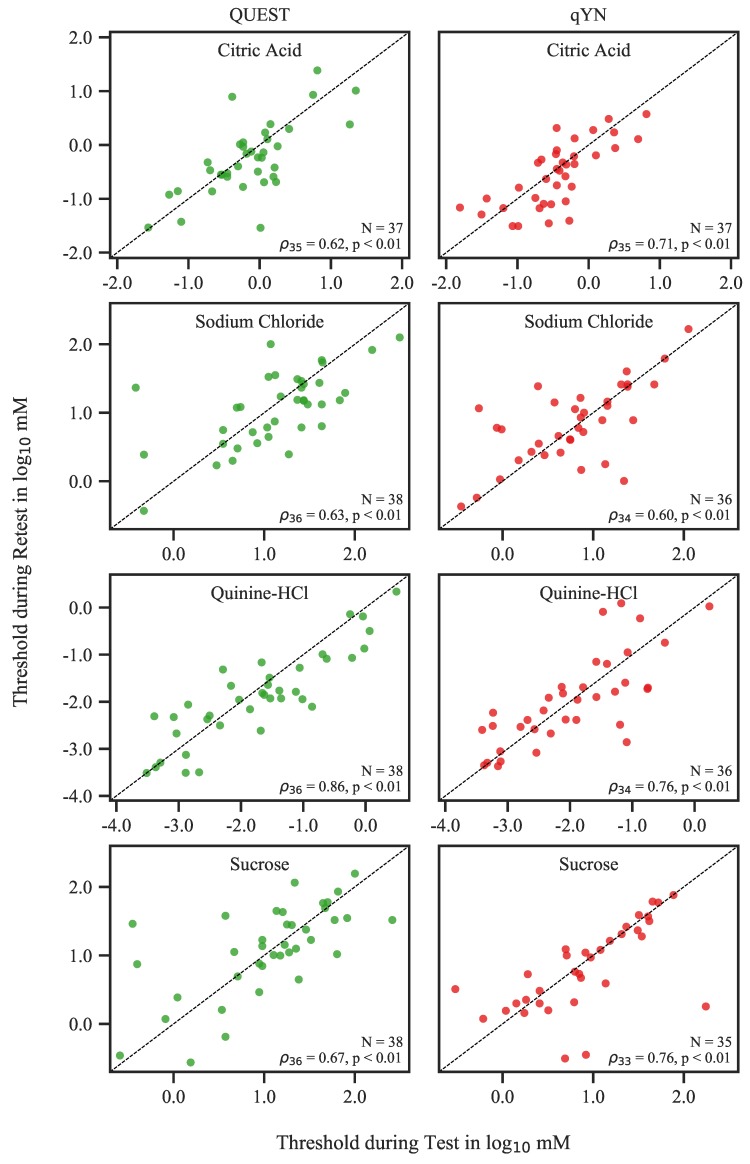
Correlation between Test and Retest threshold estimates for QUEST and qYN. Each point represents one participant; the dashed line is the identity line.

**Figure 4 nutrients-12-00024-f004:**
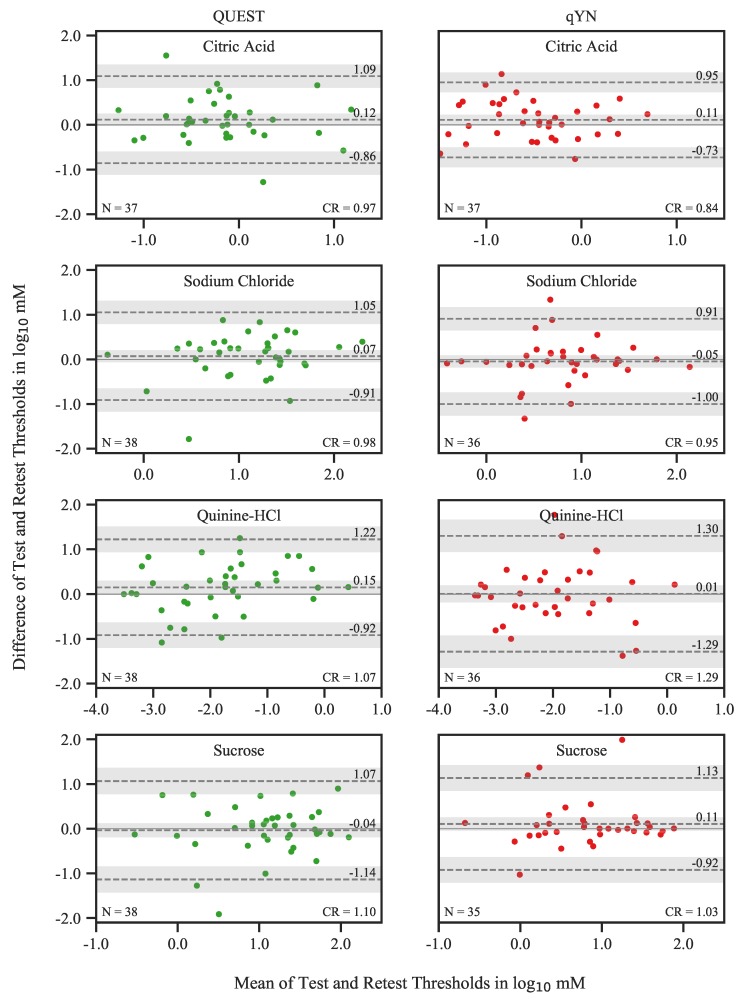
Bland–Altman plots showing differences between Test and Retest thresholds plotted over session means for QUEST and qYN. The mean difference is represented by the *dashed line in the center*, and upper and lower bounds of the limits of agreement (corresponding to the 95% CI of the differences) are shown as the *upper and lower dashed line*, respectively. The shaded areas correspond to the 95% CIs of these estimates.

**Figure 5 nutrients-12-00024-f005:**
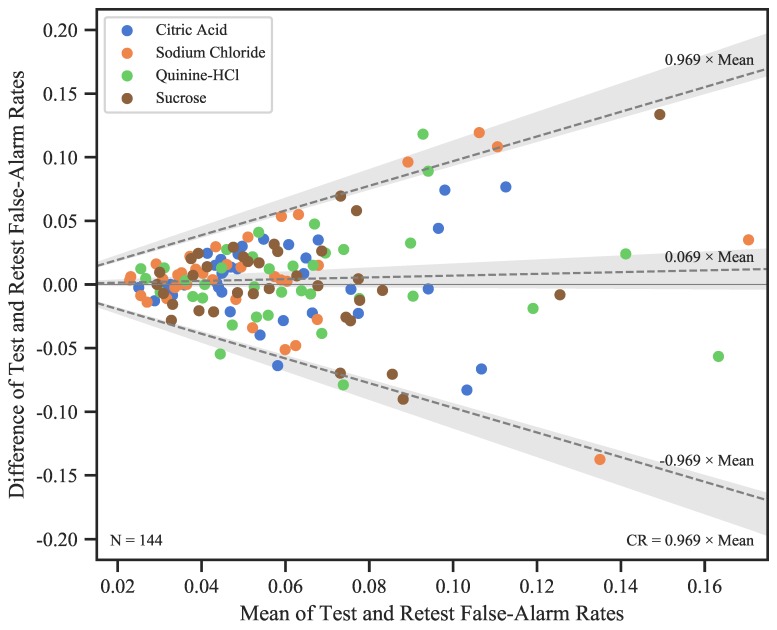
Bland–Altman plots showing differences between qYN Test and Retest FARs plotted over session means for all available Test–Retest pairs from all participants. The mean difference is represented by the *dashed line in the center*, and upper and lower bounds of the limits of agreement (corresponding to the 95% CI of the differences) are shown as the *upper and lower dashed line*, respectively. The shaded areas correspond to the 95% CIs of these estimates. Since the variability of session differences increased with session means, all calculations were performed on the $\log_{10}$-transformed data, and the results were back-transformed to their original scale for plotting. Due to the back-transformation, the dashed lines are not parallel to the abscissa, and we provide their respective formulas. Note that the intercepts of all lines were 0, and are therefore omitted.

**Figure 6 nutrients-12-00024-f006:**
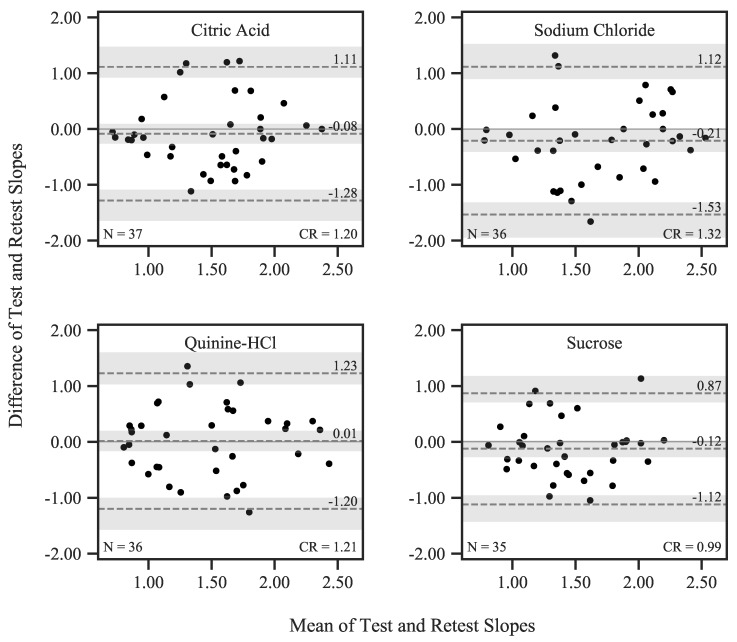
Bland–Altman plots showing differences between qYN Test and Retest $d^{\prime }$ slopes plotted over session means. The mean difference is represented by the *dashed line in the center*, and upper and lower bounds of the limits of agreement (corresponding to the 95% CI of the differences) are shown as the *upper and lower dashed line*, respectively. The shaded areas correspond to the 95% CIs of these estimates.

**Table 1 nutrients-12-00024-t001:** Questionnaire and rating data.

Measure	Mean	SD	N
DEBQ emotional eating	2.22	0.64	40
DEBQ restrained eating	2.55	0.67	40
DEBQ external eating	3.02	0.65	40
Liking of salty	3.41	1.05	37
Liking of sour	3.05	1.09	37
Liking of sweet	4.11	0.83	37
Liking of bitter	1.89	0.98	37
Consumption of salty	3.64	1.33	37
Consumption of sour	4.19	1.10	37
Consumption of sweet	4.70	1.03	37
Consumption of bitter	4.00	1.24	37

DEBQ and liking scores ranged from 1 to 5. Consumption frequencies ranged from 1 to 7.

**Table 2 nutrients-12-00024-t002:** Results of the threshold measurements during Test and Retest for QUEST and qYN.

Procedure	Tastant	N	Session	Threshold in $\log_{10}$ mM			
				mean	min	max	SD
QUEST	Citric Acid	37	Test	−0.141	−1.564	1.350	0.621
			Retest	−0.256	−1.540	1.385	0.666
	Sodium Chloride	38	Test	1.140	−0.417	2.495	0.631
			Retest	1.069	−0.432	2.100	0.544
	Quinine−HCl	38	Test	−1.737	−3.514	0.496	1.101
			Retest	−1.889	−3.514	0.339	0.953
	Sucrose	38	Test	1.054	−0.592	2.414	0.705
			Retest	1.089	−0.563	2.194	0.660
qYN	Citric Acid	37	Test	−0.446	−1.807	0.812	0.563
			Retest	−0.558	−1.508	0.574	0.598
	Sodium Chloride	36	Test	0.785	−0.457	2.052	0.607
			Retest	0.831	−0.369	2.222	0.571
	Quinine−HCl	35	Test	−1.974	−3.409	0.239	0.952
			Retest	−1.980	−3.369	0.091	0.962
	Sucrose	36	Test	0.871	−0.613	2.243	0.677
			Retest	0.765	−0.742	1.881	0.669

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
