# Peer review of "Repeatability of Taste Recognition Threshold Measurements with QUEST and Quick Yes–No"

_nutrients, 2019, doi:10.3390/nu12010024_

Round 1

Reviewer 1 Report

Reliability of taste recognition threshold measurements with QUEST and quick yes-no.

This study is focused on using a new procedure added to QUEST using a quick yes-no design to decrease false alarm rates in determining group detection thresholds for sweet, salty, and sour stimuli. Stimuli were sprayed on the anterior tongue. One concerning point is the loss of data without describing the error, leading this reviewer to be concerned about the current data analyzed. Overall, there are minor comments below.

Title: Why is QUEST and quick italicized in the title?

Abstract:

Line 2: It is unclear that the challenges listed are related to assessments of sensory perception in the clinic.

Line 7-11: Consider revision of this sentence, run-on.

Line 20-22: There is evidence supporting detection and recognition thresholds do not correspond with food preference and eating behavior.

Line 23: What does the author mean by “limited measurement duration”. Is this the time to complete the test time for participants, or length of time of surveying a participant over time?

Line 27-28: The authors reference their prior work to demonstrate that QUEST can precisely and reliably measure taste detection threshold. It would help the readers by adding additional information that this method is comparable to established thresholds using traditional methods.

Line 70: 4 should be Four.

Line 71-73: This reviewer recommends moving the participant information related to eating style to a later part, as this questionnaire has not yet been described. There is no context given for this information and why it is important to report under the participant section.

Line 74: What is the rationale for keeping data for a participant, whose questionnaire data is missing? How do you know this female was not a smoker?

Line 76: If smoking was an exclusion criteria, why are 4 participants included that are smokers?

Line 83: What is the importance of Test and Retest session capitalized?

Line 95: This reviewer recommends that the 2.2.2. header “taste ratings” be changed to Taste Preference, as ratings is often used to describe intensity ratings.

Line 97: Is this a continuous VAS scale where ratings can be placed any combination between 1 and 5, or were ratings placed on integers? This should be clearly stated how these ratings were reported.

Line 101: missing comma between sweet and sou

Line 167: Please explain why participants were required to be blindfolded.

Line 190: Why did 4 participants not complete the ratings?

Line 201: Forty one thresholds were lost due to data corruption – please explain. Because of this data corruption, how can authors be certain there were not errors in any of the other analyzed thresholds? Which thresholds were missing – this should be reported. These missing data, with out description of the error, makes this reviewer less confident in the currently reported data and analysis.

Line 258: Bitterness is disliked or less liked?  

Figure 4: Different angled triangles is not easy to observe points. Consider different shades, colors, or shapes for differences to be seen between samples.

Line 414: Taste preference or taste quality preference?

Line 420: Intake data was unrealistically low. This is concerning as these data were not reported. If they are not realistic, then the authors should describe why they analysis was conducted. Additionally, why was data with children reported as a supportive evidence when the present analysis was conducted in adults. Are there no supportive evidence in adults, if not say so.

Line 433: Test and retest is not capitalized here.

Author Response

Reviewer 1:

Note that we have highlighted all changes in the manuscript. In an attempt to improve the clarity of the paper, we have made some additional modifications (that were not specifically requested by the reviewers) in the methods and results section that we highlighted as well. These include more consistent wording and also the reorganisation of captions. 

Title: Why is QUEST and quick italicized in the title?

Reply: We used italics to indicate that these are the denominations of the methods, which may not be needed for QUEST (since it has no alternative meaning). For “quick Yes-No”, however, we wanted to insure that the method’s name cannot be mistaken. If the reviewer finds that the italization is misleading, we will remove it.

Line 2: It is unclear that the challenges listed are related to assessments of sensory perception in the clinic.

Reply: We did not intend to imply that the mentioned challenges are specific to the clinic. Rather they relate to taste sensory testing in any context. We have removed “psychological and clinical” from the sentence accordingly to increase the clarity.

Line 7-11: Consider revision of this sentence, run-on.

Reply: We have revised the sentence. It now reads:
“Variations of these parameters, however, may also influence the threshold estimate. This raises the question as to whether a procedure that simultaneously assesses threshold, false-alarm rate, and slope  might be able to produce threshold estimates with higher repeatability, i.e., smaller variation between repeated measurements of the same participant.”

Line 20-22: There is evidence supporting detection and recognition thresholds do not correspond with food preference and eating behavior.

Reply: We did not mean to disregard previous studies and have expanded the discussion on food intake and preferences (see also our reply to the next to last comment of the reviewer). We would like to mention that we have kept any reference to “food intake” brief, as we cannot add to the current body of evidence because we could not evaluate the data in our study. We felt that is would be misleading to introduce and discuss the topic in detail given that our study is largely limited to taste sensitivity. 

Line 23: What does the author mean by “limited measurement duration”. Is this the time to complete the test time for participants, or length of time of surveying a participant over time?

Reply: With “limited measurement duration” we intended to describe the time constraints that researchers and clinicians are often facing when participants or patients are examined. Since this was obviously not clear, we have rephrased it: 

“The precise and reasonably quick measurement of sensory sensitivity  has been a challenge in all senses.“

Line 27-28: The authors reference their prior work to demonstrate that QUEST can precisely and reliably measure taste detection threshold. It would help the readers by adding additional information that this method is comparable to established thresholds using traditional methods.

Reply: The reviewer touches on an important issue here. Comparison across studies is indeed difficult, as stimulus material (concentration steps), stimulus delivery (drops, spray, etc.), stimulus amount, procedure (staircase, method of limits, method of constant stimuli, etc.), and definition of “threshold” can vary and thereby influence the outcome of the threshold measurement. We have therefore elected not to compare absolute threshold values, and instead only related our test-retest correlation coefficients to those found in previous studies; this is presented in the first paragraph of the discussion section 4.2. Following the reviewer’s comment, in this section we now also provide the correlation values found in those earlier studies to facilitate the comparison between studies.

Line 70: 4 should be Four.

Reply: We have corrected this.

Line 71-73: This reviewer recommends moving the participant information related to eating style to a later part, as this questionnaire has not yet been described. There is no context given for this information and why it is important to report under the participant section.

Reply: Reflecting our motivation to measure eating style, we had included it as a mere descriptor of participants. We understand that this may be unconventional and followed the reviewer's suggestion to include the DEBQ data in a separate section (together with the ratings) in the methods and results for the DEBQ (section 2.2.2 and 3.1).

Line 74: What is the rationale for keeping data for a participant, whose questionnaire data is missing? How do you know this female was not a smoker?

Reply: The focus of the present study was on taste sensitivity measures. We therefore deem the test-retest measurements for the various tastants to be more important than the questionnaire data. Unfortunately, we cannot say whether this particular participant was a smoker. However, as we explain in our reply to the next comment, smoking was not an exclusion criterion.

Line 76: If smoking was an exclusion criteria, why are 4 participants included that are smokers?

Reply: We thank the reviewer for pointing out the discrepancy. Smoking was not an exclusion criterion in this study. We regret this oversight and corrected the text accordingly.

Line 83: What is the importance of Test and Retest session capitalized?

Reply: We capitalized Test and Retest to highlight that we are referring to the respective (initial and repeated) sessions and not using the verb/noun. 

Line 95: This reviewer recommends that the 2.2.2. header “taste ratings” be changed to Taste Preference, as ratings is often used to describe intensity ratings.

Reply: We have changed the header accordingly.

Line 97: Is this a continuous VAS scale where ratings can be placed any combination between 1 and 5, or were ratings placed on integers? This should be clearly stated how these ratings were reported.

Reply: The ratings were given as integers. We have specified how the ratings were given, which is on a 5-point-Likert scale (which was only mentioned in the data file that we uploaded to Zenodo). It now reads als in the manuscript:

“...horizontal 5-point Likert scale anchored with 1 (not at all) and 5 (extremely)...”

Line 101: missing comma between sweet and sour

Reply: We have added the comma.

Line 167: Please explain why participants were required to be blindfolded.

Reply: Blindfolding has been shown to reduce distraction and thereby help participants focus on the taste. Additionally, it ensures that participants cannot see which stimulus bottle is being presented. It is also commonly used in smell testing. We have added the rationale to the methods: 

“Participants were seated comfortably and blindfolded to reduce distraction and improve focus.”

Line 190: Why did 4 participants not complete the ratings?

Reply: As also detailed in our reply to the next comment, we have suffered some loss of data during the transition of the lab from one research institute to another. This, unfortunately, included also the taste ratings of four participants and for one of them also the DEBQ. While this is most undesirable and certainly not how we typically operate, we felt strongly about including this information in the manuscript. And because the questionnaires were a comparably small part of the data collected in the present study, we kept those participants in the sample. We would like to mention that all missing data was disclosed additionally in the data spreadsheet that we had uploaded to Zenodo and referenced in the paper. This spreadsheet will remain available for the readers. 

Line 201: Forty one thresholds were lost due to data corruption – please explain. Because of this data corruption, how can authors be certain there were not errors in any of the other analyzed thresholds? Which thresholds were missing – this should be reported. These missing data, without description of the error, makes this reviewer less confident in the currently reported data and analysis.

Reply: The reviewer raises an important point that we will gladly explain in more detail. All threshold data were properly collected and stored on a computer during testing. Their integrity (completeness of files and plausibility of results) was confirmed by the experimenter immediately after each test session. “Data corruption” (loss of some of the files) only happened when the electronic data was transferred during the transition of the research lab from Potsdam to Jülich. We are, therefore, absolutely confident about the integrity of the data reported here. Since the term “data corruption” may be misleading, we have clarified it: “... were lost during the transfer of electronic data”. We would like to mention that all missing data was disclosed in the data spreadsheet that we had uploaded to Zenodo. We have now updated it such that the reported (complete) and the partial (incomplete) taste thresholds are listed in separate tabs to facilitate evaluation of the data for the readers. 

Line 258: Bitterness is disliked or less liked?  

Reply: While bitter is “more disliked” than the other tastes, we used the term “less liked” in reference to the scale of “liking” for the direct comparison of the four different tastes in the results section. We have, however, added an additional  explanation / interpretation of the ratings at the end of the paragraph: 

“In contrast to sweet, sour, and salty, which were generally liked (as indicated by ratings larger than 2.5), bitter was clearly disliked (as indicated by a rating smaller than 2.5).”

Figure 4: Different angled triangles is not easy to observe points. Consider different shades, colors, or shapes for differences to be seen between samples.

Reply: We absolutely agree and have recreated figure 4 using different colors for the different taste qualities.

Line 414: Taste preference or taste quality preference?

Reply: We have modified the singular occurrence of “taste quality preference” to “taste preference”, which is now consistently used throughout the paper.

Line 420: Intake data was unrealistically low. This is concerning as these data were not reported. If they are not realistic, then the authors should describe why they analysis was conducted. 

Reply: We merely reported intake data for reasons of transparency. We agree with the reviewer that it makes sense to not perform any statistical analyses here. We have therefore removed the ANOVA and simply report the means and SDs:

“Participants reported to consume food and beverages of the different tastes only approximately once per week (score=4) for all taste qualities (see table 1). Because the reported mean frequencies raise doubts as to their validity, no further statistical analyses were conducted.”

Additionally, why was data with children reported as a supportive evidence when the present analysis was conducted in adults. Are there no supportive evidence in adults, if not say so.

Reply: We had referenced the study by Sauer et al. because it has shown a link between sour taste ability, preference, and food intake, which fits with our observation of a link between sour taste preference and thresholds. We have now added additional references to studies in adults and a more general conclusion on the topic: 

“A recent study, however, in which food and beverage consumption was thoroughly assessed (Cattaneo2019), showed clear associations between salty, sweet, and bitter taste sensitivity and the intake of foods and beverages with these taste qualities. In this study, higher sensitivity (low thresholds) was linked with lower food intake (or vice versa). No such link was found for sour taste, though, leaving it open whether taste sensitivity should be considered a general predictor for food intake behavior. The observation that a low sour taste preference and poor sour taste abilities improved through the course of a weight loss intervention obese children, indicates, however, that dietary changes may influence preference as well as taste function, at least to some extent (Sauer2017). The latter data provide a glimpse into the complex and potentially reciprocal interplay of taste function, taste preferences, and food intake and a review on the determinants of fruit and vegetable consumption revealed that other factors such as age, gender, socio-economic status, preferences, parental intake, and  availability play a major role as well (Rasmussen2006), thereby corroborating the multifaceted nature of food intake behavior and highlighting the need for further studies.”

Line 433: Test and retest is not capitalized here.

Reply: We have now capitalized Test and Retest (when referring to the specific sessions in our experiments) throughout the manuscript and kept the term “test-retest correlation” as it cannot be easily mistaken.

Reviewer 2 Report

This is a methodological paper comparing 2 psychophysical procedures for repeatability.  There was no difference in the repeatability of the measures.  Thresholds appeared higher using Quest but Quest took less time per participant. 

As an overall comment I think the paper is reasonably accessible considering the content.  Transitions to introduce the reader to the plots and statistics are useful. A table defining all of the parameters of interest for quick reference would help though it does become a bit of an alphabet soup. 

Figure 4- Please use symbols that are easier to distinguish between the taste stimuli. Also in Fig. 4 there is an n of 144 listed, I assume this represents repeated measures from individuals? I wonder about measures then being autocorrelated i.e. there were participants that were very bad in all stimuli? As suggested by the text in 4.3.

Can you clarify that the same individuals in Fig 5 sum to be the samples in Figure 4? 

Section 6 suggests that graphical representations of individual thresholds are available.  I did not find them.  I do think there would be value adding representative figures to the paper so that differences in slope and retest could be seen in the function which may be more intuitive to look at.

Author Response

Reviewer 2:

Note that we have highlighted all changes in the manuscript. In an attempt to improve the clarity of the paper, we have made some additional modifications (that were not specifically requested by the reviewers) in the methods and results section that we highlighted as well. These include more consistent wording and also the reorganisation of captions. 

As an overall comment I think the paper is reasonably accessible considering the content.  Transitions to introduce the reader to the plots and statistics are useful. A table defining all of the parameters of interest for quick reference would help though it does become a bit of an alphabet soup. 

Reply: We have included a table (now table 1) with the means and SDs for all ratings and questionnaires, and removed the values from the text to improve clarity. The threshold results can be found in table 2 (former table 1). 

Figure 4- Please use symbols that are easier to distinguish between the taste stimuli. 

Reply: We agree with the reviewer that the previously used triangular markers can be difficult to distinguish. We have now replaced them with colored dots.

Also in Fig. 4 there is an n of 144 listed, I assume this represents repeated measures from individuals? 

Reply: N is indeed the number of Test-Retest pairs that have entered analysis. We have adjusted the caption of the figure (now Fig. 5) to now read 

“... Markers show differences between Test and Retest FARs for all available Test-Retest pairs from all 37 participants.”

I wonder about measures then being autocorrelated i.e. there were participants that were very bad in all stimuli? As suggested by the text in 4.3.

Reply: We found indeed no systematic difference between tastes in false-alarm rate (FAR) (see new section 3.2.2: no main effect of “tastant” or “session”), i.e., a participant with a low FAR for one taste would likely also have low FAR for the other tastes. This is also why we present the data for all tastants in a single plot (and not in four separate plots) as Fig. 5.

Spearman correlations between Test and Retest FARs (i.e., across all tastants), assessed for each participant individually, show a monotonic relationship of a medium strength (mean absolute correlation coefficient of 0.56). This result, together with the ANOVA mentioned above, suggests that participants were pretty consistent in their response behavior between the different threshold runs, which is highly desirable. 

We also calculated the absolute differences between Test and Retest FARs and averaged the results across tastants within each participant to assess mean absolute FAR differences (see box plot attached to this reply).

It can be seen that the mean absolute FAR differences between both sessions were below ~0.05 for all but two participants. This result, again, suggests a quite consistent response behavior within individual participants.

Can you clarify that the same individuals in Fig 5 sum to be the samples in Figure 4? 

Reply: Indeed, the total number of individuals (or Test-Retest pairs) in both figures is identical.

We have changed the caption of the previous Fig. 4 (now Fig. 5)  to make it more obvious that all measurements of all participants are shown here, as stated in the response above:

“... Markers show differences between Test and Retest FARs for all available Test-Retest pairs from all 37 participants.”

Section 6 suggests that graphical representations of individual thresholds are available. I did not find them. I do think there would be value adding representative figures to the paper so that differences in slope and retest could be seen in the function which may be more intuitive to look at.

Reply: We would like to apologize for accidentally not including the promised figures in the data repository. This is now resolved, and not only all data but also all figures can be retrieved from https://doi.org/10.5281/zenodo.3540534. To facilitate evaluation of the link between parameters, we have now included a new Figure 1 that displays the mean psychometric functions for test and retest and both procedures for all tastants.
